# The Regulatory Mechanism of miR-574-5p Expression in Cancer

**DOI:** 10.3390/biom13010040

**Published:** 2022-12-26

**Authors:** Wei Huang, Yifan Zhao, Zhengyi Xu, Xiaoyue Wu, Mingxin Qiao, Zhou Zhu, Zhihe Zhao

**Affiliations:** State Key Laboratory of Oral Diseases & National Clinical Research Center for Oral Diseases, West China Hospital of Stomatology, Sichuan University, Chengdu 610041, China

**Keywords:** miR-574-5p, cancer, diagnostic and prognostic markers, non-coding RNA network

## Abstract

MicroRNAs (miRNAs) are a group of small, single-stranded, non-coding RNAs approximately 22 nucleotides in length. The dysregulation of miRNAs has been widely investigated in various pathological processes, including tumorigenesis, providing a biomarker for cancer diagnosis and prognosis. As a member of the miRNA family, miR-574-5p is located on the human chromosome 4p14 and is highly correlated with a high incidence of human cancers. Functional pathways as well as underlying novel mechanisms upregulate or downregulate miR-574-5p, which plays an important regulatory role in tumorigenesis and progression. In this review, we systematically summarize the context-dependent implications of miR-574-5p and review differences in miR-574-5p expression in cancer. We also investigate the intricate functions exerted by miR-574-5p in diverse pathological processes and highlight regulatory pathways, networks, and other underlying novel mechanisms. The clinical applications of miR-574-5p as a diagnostic biomarker, prognostic biomarker, and therapeutic mechanism are also discussed in this paper. On this basis, we anticipate that miR-574-5p will be a promising and effective biomarker and therapeutic target.

## 1. Introduction

MicroRNAs (miRNAs) are a group of small, single-stranded, non-coding RNAs approximately 22 nucleotides in length [1]. Since firstly discovered in the nematode *C. elegans*. in 1993 [2], thousands of miRNA genes have been annotated in diverse species, including humans. Either residing in the intergenic or intragenic regions of the genome, miRNAs account for approximately 1-5% of the human genome and regulate over 30% of protein-coding, survival-related genes based on predictions [3,4]. In recent years, a large body of literature has accrued regarding their critical role in post-transcriptionally regulating gene expression, mainly acting through binding to the 3’-untranslated regions (UTRs) of messenger RNAs (mRNAs). According to the different degree and nature of complementarity between miRNA sequences and the target mRNA 3’UTRs, the latter translation could be completely degraded or partially inhibited, resulting in delicate modulations of a variety of fundamental cellular and physiological activities, such as cell proliferation, embryogenesis, and development [5,6]. With advances in experimental and computational methodologies in the miRNA research field, the dysregulation of miRNAs has been widely investigated in various pathological processes, including tumorigenesis, rendering them a promising diagnostic and therapeutical target in the future [7]. 

Recently, a growing number of studies have highlighted the significant role of miR-574-5p, a member of the miRNA family, in multiple human diseases. Human miR-574-5p (has-miR-574-5p) is encoded by mir574 (gene ID: 693159), which is located on the human chromosome 4p14 and is generated in a multi-step process (Figure 1). Transcription begins with RNA polymerase II, which produces an initial long transcription chain called pri-miRNA. Unlike the canonical miR biogenesis pathway, pri-miRNAs can be spliced and debranched into pre-miRNA hairpins that are suitable for Dicer cleavage, thus bypassing the microprocessor [8,9,10]. Upon being exported to the cytoplasm by exportin-5 (XPO5), pre-miRNAs are cleaved by two RNases and turned into an RNA duplex of approximately 22 nucleotides [11]. In the case of miR-574-5p, the strand originating from the 5’ side of the pre-miRNA (thus annotated with the suffix -5p) is selectively loaded on the argonaute protein (AGO) to form the miRNA-induced silencing complex (RISC) [12]. Unlike the usually degraded undominated strand, the 3’ strand of miR-574 still gives rise to the functional miR-574-3p counterpart, and the 5p/3p ratio reportedly contributes to some pathologies [13,14]. 

In this review, we discuss the context-dependent implications of miR-574-5p in a variety of diseases, including some of the most fatal cancers and other diseases, such as cardiovascular and neurological problems. Additionally, as miRNAs have been recognized as a control hub or node in the intricate regulatory network, we focus on how it works and is regulated, highlighting the downstream targets and functional pathways of miR-574-5p dysregulation in cancer, as well as the typical non-coding RNA network. Some novel findings of its regulatory mechanism in cancer are also noted in an attempt to fuel our understanding of the underlying mechanisms of miRNA functions and regulations. In addition, although still in infancy, the promising clinical potential of miR-574-5p also deserves attention, as emerging studies have shed light on its value as a biomarker and therapeutic target.

## 2. The Context-Dependent Role of miR-574-5p in Cancer

Dysregulated miR-574-5p has been found in all pathological processes of tumorigenesis, including cancer cell proliferation, migration, invasion, metastasis, apoptosis, epithelial-mesenchymal transition, and angiogenesis (Figure 2). However, numerous studies have found that the role of miR-574-5p is significant but also controversial, acting as both a tumor suppressor and oncogene according to the tumor type and associated pathways. Therefore, its implication in cancer might be context-specific, which we further discuss below. 

### 2.1. Lung Cancer

With an estimated 1.8 million deaths worldwide, lung cancer is the leading cause of cancer-related death, remaining one of the most severe global health issues [15,16]. Histologically, lung cancer can generally be divided into two major classes: small cell lung carcinoma (SCLC) and non-small cell lung carcinoma (NSCLC). The latter, which accounts for approximately 85% of all cases, includes subtypes such as lung adenocarcinoma (LUAD) and lung squamous cell carcinoma (LUSC) [17]. Despite all the clinical progress during these years, the molecular heterogeneity, high relapse rate, and overall low survival rate of lung cancer continue to spur research into the underlying pathological mechanisms, diagnosis, and therapies, where miR-574-5p might play a role.

Early in 2012, a study by Li. et al. characterized the oncogenic role of miR-574-5p in lung cancer [18]. miRNA array analysis showed that it was the most upregulated miRNA in human lung cancer cells under Toll-like receptor 9 (TLR9) signaling. Experimental downregulation of miR-574-5p both in vitro and in vivo inhibited enhanced tumor progression, further validating its involvement in lung cancer. As the contradictory functions of miR-574-5p have been evidenced by a growing number of studies this year, it may be speculated that miR-574-5p plays a context-dependent role on different types of lung cancer, both promoting and inhibiting oncogenesis.

On the one hand, the majority of studies indicated that miR-574-5p was a promoter of NSCLC development. Research by Zhou et al. indicated that miR-574-5p was involved in the progression and metastasis of NSCLC, as it enhanced the migration and invasion of cancer cells mainly through inhibiting the expression of its downstream target, protein tyrosine phosphatase receptor type U (PTPRU) [19]. Similarly, Rho family GTPase 3 (RND3) has also been recognized as a downstream target of miR-574-5p, the inhibition of which significantly accelerated apoptosis and suppressed adenocarcinoma progression [20]. In addition, recent studies have noted that miR-574-5p-activated prostaglandin E2 (PGE2) synthesis, a crucial inflammatory lipid mediator, was also related to NSCLC development in an intricate manner [21]. 

On the other hand, studies with SCLC, a less predominant but also significant subtype of lung cancer, mostly reported miR-574-5p as a tumor suppressor through the non-coding RNA network. As indicated in two studies, miR-574-5p partially exerted a protective effect as a long non-coding RNA (lncRNA) sponge, antagonizing the oncogenic lncRNA HOTTIP (HOXA transcript at the distal tip) and affecting the expression of downstream targets, such as enhancer of zeste homolog 1 (EZH1) protein and vimentin (VIM), ultimately alleviating SCLC cell growth, proliferation, and also EMT [22,23]. However, as another study showed miR-574-5p facilitated metastasis of SCLC [24], more research is required.

### 2.2. Thyroid Cancer

Thyroid cancer, the most common malignant tumor of the endocrine system, includes papillary thyroid carcinoma (PTC), follicular thyroid carcinoma (FTC), anaplastic thyroid carcinoma (ATC), and medullary thyroid carcinoma (MTC), with PTC being predominant among all subtypes [25,26]. Although most cases of PTC have a better prognosis than other types of malignant tumors, cervical lymph node metastasis can occur at the beginning of diagnosis and severely influences the 10-year survival rate [27,28]. Other subtypes, such as FTC, despite accounting for a smaller proportion of thyroid cancer cases, has higher distant metastasis rates and worse prognosis [29]. Therefore, it is of great clinical significance to understand the underlying pathogenic mechanism of thyroid cancer, where miR-574-5p serves as a candidate oncogene.

In 2013, a study found that papillary thyroid carcinoma susceptibility candidate 3 (PTCSC3), a newly identified lncRNA, could selectively bind with miR-574-5p. Significant growth inhibition, cell cycle arrest, and increased apoptosis were observed in three pathological types of thyroid cancer cells with PTCSC transfection [30]. Following this axis, another study further confirmed that PTCSC/miR-574-5p mediated the PTC-1 cell cycle through targeting suppressor of cancer cell invasion (SCAI), thus promoting the proliferation and migration of PTC via Wnt/β-catenin pathway [31]. In addition to SCAI, Zhang et al. demonstrated that the disrupted cell cycle and apoptosis induced by miR-574-5p were associated with another target, quaking proteins (QKIs) [32]. Additionally, fork head box N3 (FOXN3), a key cell cycle regulator and transcriptional tumorigenesis suppressor, was also identified as a direct target of miR-574-5p in thyroid cancer progression [33,34]. Conclusively, the oncogenic function of miR-574-5p in thyroid cancer was closely related to the cell cycle and Wnt/β-Catenin signaling, and novel targets remain to be identified.

### 2.3. Breast Cancer (BC)

Breast cancer is the most common cancer type in women, and female breast cancer has already surpassed lung cancer as the most diagnosed cancer, with an estimated 2.3 million new cases (11.7%) according to statistics in 2020 [15]. As a heterogeneous disease, breast cancer can be divided into 3 major subtypes in terms of molecular markers for estrogen or progesterone receptors and human epidermal growth factor 2 (ERBB2; formerly HER2) [35]. Triple-negative BC (TNBC), defined as the absence of all three molecular targets, accounts for approximately 15% of BC cases and relates to worse therapeutic response to chemotherapy or hormone therapy, as well as a higher risk of distant relapse and poorer prognosis compared with other types [36,37,38]. However, the underlying pathogenic mechanisms of TNBC remain elusive. 

In recent years, the protective role of miR-574-5p in breast cancer, especially TNBC, has been gradually brought to light. A study by Zhang et al. reported that miR-574-5p could repress TNBC cell proliferation, migration, and epithelial-mesenchymal transition (EMT) in vitro while reducing tumor growth and metastasis in vivo. It was demonstrated that miR-574-5p could simultaneously target SRY (sex-determining region Y)-box 2 (SOX2), B-cell lymphoma/leukemia 11A (BCL11A), and thus inhibit the SKIL/transcriptional co-activator with PDZ-binding motif (TAZ)/connective tissue growth factor (CTGF) axis to regulate malignant phenotypes in TNBC [39]. 

As TNBC tends to metastasize distantly to specific organs, such as the lungs, a high level of miR-574-5p inhibited the migration and invasion abilities in lung-metastatic TNBC cells, whereas its knockdown resulted in enhanced metastasis performance [40,41]. Further experiments demonstrated that miR-574-5p was the shared miRNA for the upstream linc-ZNF469-3 and downstream target ZEB1, highlighting the role of linc-ZNF469-3/miR-574-5p/ZEB1 in TNBC lung metastasis.

### 2.4. Gastric Cancer (GC)

As the third leading cause of cancer-related deaths worldwide, gastric cancer represents a huge clinical challenge due to its aggressive characteristics and poor prognosis [42]. Although relatively fewer studies have focused on the role of miR-574-5p in gastric cancer, the existing literature has already shed light on some novel mechanisms of miRNAs pathogenic functions. 

For instance, a study in 2020 first elucidated that miR-574-5p engaged in gastric cancer by promoting angiogenesis, a crucial pathological process for tumor growth, development, and metastasis [43,44,45]. It was found that miR-574-5p was upregulated in response to hypoxic conditions both in vitro and in vivo, and inhibition of miR-574-5p resulted in decreased endothelial growth factor A (VEGFA) expression, as well as reduced viability, migration, invasion, and tube formation of human umbilical vein endothelial cell lines (HUVECs). Unexpectedly, this promotion was achieved through activation of 44/42 mitogen-activated protein kinases (MAPKs) by mir574-targeted inhibition of protein tyrosine phosphatase non-receptor type 3 (PTPN3), revealing another way to achieve VEGFA regulation besides hypoxia-inducible factor-1α (HIF-1α) stability [46]. 

In addition, as miR-574-3p, the counterpart of miR-574-5p, also contributed to angiogenesis in gastric cancer development by targeting cullin2, the aberrant ratio of 5p/3p is another critical player in gastric cancer, as revealed by Zhang et al. in 2019 [13,43]. Their findings indicated that although derived from the same precursor, miR-574-5p and 3p arms were inversely expressed in patients and exerted opposite effects on gastric carcinogenesis with different downstream targets, QKI6 and ACVR1B, respectively. Based on the target RNA directed miRNA degradation (TDMD) theory, these authors revealed that the upregulated 5p/3p ratio, namely miR-574 arm-imbalance, was partially attributed to the dynamic expression of the highly complementary targets of miRNAs, and in turn exacerbated gastric cancer progression, correlating with poorer prognosis with advanced TNM stages.

In addition, as environmental carcinogens are an important etiologic factor in gastric carcinogenesis, miR-574-5p was also shown to be involved in environmental carcinogen-induced gastric cancer by interfering with tumor-suppressive lncRNA LOC101927497, thus promoting the proliferation and migration of N-methyl-N′-nitro-N-nitrosoguanidine (MNNG)-induced malignantly transformed human gastric epithelial cells [47,48].

### 2.5. Colorectal Cancer (CRC)

Although CRC remains the second most common cause of cancer death in the United States, advancements in screening and detection techniques, as well as the understanding of risk factors, pathogenesis, and precursor lesions, have resulted in the reduction of morbidity and mortality of CRC [49,50]. Intriguingly, previous reports about miR-574-5p involvement in CRC demonstrated controversial results.

In 2013, Ji et al. first unraveled the oncogenic role of miR-574-5p in CRC. In CRC tissue from both C57BL/6-Apcmin+ mice and clinical patients, the expression of miR-574-5p was shown to be significantly upregulated, which was inversely correlated with the Quaking family of RNA-binding proteins (Qki) and positively correlated with β-catenin [51]. Further experiments confirmed this functional axis, revealing that miR-574-5p could negatively regulate Qki6/7 via interactions with miRNA recognition elements (MREs) on the corresponding 3’UTRs, thus exerting oncogenic effects such as increased CRC cell proliferation, invasion, and migration, decreased differentiation, and cell cycle exit. These oncogenic effects were further investigated in another study in 2019, which indicated that miR-574-5p accelerated the migration and invasion of CRC cells by targeting downstream calcium-binding and coiled-coil domain 1 (CALCOCO1), whereas the non-coding RNA LINC00052 inhibited CRC metastasis via sponging miR-574-5p and modulating CALCOCO1 expression [52]. 

However, a contradictory conclusion was drawn in another report, demonstrating that hsa-miR-574-5p played a suppressive role in colorectal cancer liver metastasis by negatively directing MACC-1 expression [53]. In addition, different lncRNA networks may act divergently in the same disease. Li et al. reported the protective effects of miR-574-5p for CRC development through regulation of the lncRNA MFI2-AS1/miR-574-5p/MYCBP axis [54]. Conclusively, the complexity of the miR-574-5p regulatory network calls for greater exploration and clearer elucidation in this field (Table 1).

### 2.6. Others

There is growing evidence recognizing miR-574-5p as a potential oncogene in esophageal squamous cell carcinoma (ESCC). In 2020, a study by Guo et al. found that downregulated miR-574-5p could inhibit ESCC cell proliferation and promote apoptosis via directly targeting Zinc finger protein 70 (ZNF70), and these functions were correlated with reactive oxygen species (ROS) generation and MAPK pathways [55]. Unregulated miR-574-5p has been found in ESCC tumor tissues according to other studies as well [56,57].

In addition, the tumor-promotive role of miR-574-5p has been revealed in other types of cancer, such as cervical cancer and nasopharyngeal carcinoma [31,33], respectively targeting QKI and FOXN3, with which interactions were previously discussed in gastric cancer, CRC, and thyroid cancer [13]. Conversely, some reports have identified miR-574-5p as a tumor-suppressor in some relatively rare tumors, such as glioma and chordoma. 

In conclusion, given disease-context effects, numerous functional targets and pathways, intricate regulatory networks, and elusive molecular mechanisms of miR-574-5p are involved in cancer incidence, progression, and metastasis, and more in-depth studies are needed in the future (Table 2).

## 3. The Functional Pathways and Regulatory Mechanisms of miR-574 in Cancer

As discussed above, a plethora of studies have indicated the involvement of miR-574-5p in a variety of cancers through targeting and interacting with diverse targets (Figure 3). Here, we specifically summarize some of the most extensively studied functional pathways of miR-574-5p in cancer in order to fuel our understanding the role of miRNAs in tumorigenesis and progression.

### 3.1. Typical Functional Pathways of miR-574-5p in Cancer

#### 3.1.1. Wnt/β-Catenin Pathway

The Wnt/β-catenin signaling pathway, as the canonical β-catenin-dependent Wnt signaling pathway, refers to a family of conserved signaling proteins that are involved in many basic physiological processes, such as cell survival, proliferation, apoptosis, and tissue homeostasis. It was demonstrated that dysregulation of the Wnt/β-catenin contributes to cancer development and progression, with accumulating evidence shedding light on its activation in miR-574-5p-promoted tumorigenesis. Based on current research, as some protein-coding RNAs targeted by miR-574-5p, such as QKI, PTPRU, SCAI, and FOXN3, are negative regulators of the Wnt/β-catenin pathway, miR-574-5p overexpression is capable of suppressing these proteins and activating β-catenin signaling in diverse cancer types, including SCLC, NSCLC, CRC, thyroid cancer, and cervical cancer.

#### 3.1.2. MAPK Pathway

In addition to the extensively investigated Wnt pathway, some latest studies also identified the mitogen-activated protein kinase (MAPK) pathway as a player in miR-574-5p’s tumorigenic function. As a series of protein Ser/Thr kinases that perform transmembrane signal transduction, the MAPK pathway regulates diverse basic cellular functions in almost all eukaryotic cells [58,59]. As previously mentioned, through directly inhibiting PTPN3, a protein tyrosine phosphorylase capable of removing the phosphate group on the phosphorylated tyrosine residue of kinases, miR-574-5p promoted angiogenesis and tumorigenesis in gastric cancer with an increased level of phosphorylation of 44/42 MAPKs [43,60]. However, a 2020 report revealed that miR-574-5p was a contributor to ESCC development via inhibition of the MAPK pathway [55]. It was found that downregulating miR-574-5p could induce the activation of JNK, ERK, and p38 MAPK in a ROS-dependent manner, as MAPKs serve as one of the major ROS-sensitive signal-transducing pathways and upstream signals for apoptosis initiation [61,62]. In consideration of the complexity of the MAPK pathway, more in-depth investigations of the MAPK pathway in miR-574-5p-regulated tumorigenesis are required.

### 3.2. Other Novel Mechanisms of miR-574-5p Regulation in Tumorigenesis

#### 3.2.1. Arm-Imbalance

In addition to the functional pathways and ceRNA network mentioned above, emerging literature has provided some advanced insight into the underlying mechanisms of miR-574-5p in recent years. Among them, the arm-imbalance between miR-574-5p and 3p is a novel mechanism that underlies the progression of gastric cancer. It was found that although derived from the same precursor miR-574, the two strands played reverse roles in GC development with distinct downstream targets, and upregulation of the 5p/3p ratio was correlated with aggressive disease stage [13]. 

One possible theoretical explanation for the imbalanced 5p/3p ratio in gastric carcinogenesis is the target RNA-directed miRNA degradation (TDMD), which means the highly complementary targeted RNAs of miRNAs could trigger their degradation [63,64]. As validated by various experimental methods in the report, several 5p-targeted tumor-suppressive mRNAs were downregulated during carcinogenesis, thus increasing the miR-574-5p level, whereas the miR-574-3p level decreased when its target level were raised. The broken target-miRNA homeostasis in GC exerted positive feedback, as upregulated miR-574-5p further inhibited more tumor suppressive targets while the decreased counterpart 3p promoted oncogenes, synergistically accelerating GC progression with more slanted imbalance. With this finding in mind, instead of merely focusing on miR-574-5p, the maintenance and re-modification of miR-574-targeted homeostasis may be of greater significance for cancer treatments in the future.

#### 3.2.2. Origin-Dependent Role of miR-574-5p

Another interesting finding is that the subcellular distribution of miR-574-5p may also serve as a critical regulator for miRNA functions in pathogenesis. Specifically, Saul et al. revealed in 2019 that miR-574-5p could act as an RNA decoy to CUG RNA-binding protein 1 (CUGBP1), thus inducing microsomal prostaglandin E synthase-1 (mPGES-1) expression with enhanced alternative splicing and generation of an mPGES-1 3’UTR isoform, finally promoting tumor growth in vivo [65,66]. However, a recent study in 2021 contradicted this conclusion, suggesting the effects were only related to intracellular miR-574-5p, whereas those derived from small extracellular vesicles (sEV) exhibited the opposite effects, thus downregulating mPGES-1 and the PGE2 level. The seemingly contradictory effect depends on miRNA transfer via sEV to the novel subcellular location, enabling sEV-derived miR-574-5p to activate endosomal Toll-like receptors (TLR) 7/8 in a cell-specific behavior that is uniquely limited to adenocarcinoma [67]. They also identified that the combination of intracellular and sEV-derived miR-574-5p delicately regulated PGE2-levels through a feedback loop since inflammatory PGE2 induced by the tumor environment could trigger miR-574-5p sorting into sEV and upregulate the sEV-derived miR-574-5p level. Conclusively, it was found that the same miRs could exert distinct functions in tumorigenesis based on their subcellular localization for the first time.

## 4. miR-574-5p in Other Diseases

Significantly, despite most attention having been focused on the role of miR-574-5p in cancer, its implication in pathogenesis is far beyond only cancer, involving a wide range of other non-cancerous diseases of different systems. 

For example, several studies discussed the dual effects of miR-574-5p in cardiovascular disease. It was found that miR-574-5p expression was upregulated in both the sera and vascular smooth muscle cells (VSMCs) of coronary artery disease (CAD) patients, and that overexpression promoted cell proliferation and inhibited apoptosis by directly targeting the ZDHHC14 gene. As the aberrant proliferation of VSMCs contributes to atherosclerotic plaque formation, a salient pathogenic factor of CAD, miR-574-5p was proven as a CAD-related factor involved in disease development [68]. Additionally, another study revealed the correlation between miR-574-5p and cardiac fibrosis after myocardial infarction (MI). Through binding to the 3’UTR of ARID3A, miR-574-5p promoted the fibroblast-to-myofibroblast conversion of human cardiac fibroblasts (HCFs) attributed to TGF-β-induced cardiac fibrosis [69]. 

Moreover, in addition to serving as a pathogenic factor, miR-574-5p also exerts cardioprotective effect in the heart. A study by Wu et al. demonstrated that both 5p and 3p strands could be induced under pathogenic cardiac stress [70]. The experiments showed that genetic knockout or exogenous injection of miR-574 significantly influenced the disease phenotype, resulting in exacerbated or attenuated pathological cardiac remodeling, respectively. Mechanistically, the protective effect depended on its downstream target FAM210A (family with sequence similarity 210 member A), which interacted with mitochondrial translation elongation factor EF-Tu and modulated mitochondrial-encoded protein expression, a key player in normal cardiac functions [71].

Similarly, the protective role of miR-574-5p was also investigated in other systems, such as the respiratory [72], urinary [73], digestive [74,75], and nervous systems [76]. For instance, a study in 2020 illustrated that, through targeting HMGB1 (high-mobility group box-1), a key cytokine that mediates the response to inflammation, injury, and infection, miR-574-5p evidently alleviated acute respiratory distress syndrome (ARDS) [72]. In ARDS context, the miR-574-5p level was upregulated upon LPS stimulation via TLR/NF-κB–dependent pathways, whereas overexpression of miR-574-5p suppressed LPS-induced inflammatory responses and inhibited the activation of NF-kB signaling and the NLRP3 inflammasome through HMGB1 inhibition. Further, researchers found that miR-574-5p could protect against neuroinflammation, as overexpressed miR-574-5p in the hippocampal region could decrease the targeted BACE1 expression, and therefore restored synaptic function, improved spatial memory, and learning impaired by PM2.5 exposure.

Finally, the role of miR-574-5p as a pathogenic factor in some less discussed diseases should not be overlooked. For instance, two studies focused on the implication of miR-574-5p in preeclampsia, a hypertensive pregnancy disorder that affects 3–5% of pregnancies and is characterized by high blood pressure, proteinuria, and maternal organ dysfunction [77]. It was revealed that overexpression of miR-574-5p in preeclampsia endothelial cells significantly weakened the endothelial wound healing capacity through inhibition of downstream expression of SLC31A1, a key encoding gene for membrane copper transport. In this way, aberrantly upregulated miR-574-5p disrupted endothelial cell proliferation and migration by limiting copper transportation, which may contribute to the antiangiogenic environment in preeclampsia. Moreover, another study found that miR-574-5p contributed to preeclampsia development through interaction with the lncRNA AGAP2-AS1 [78]. Growing evidence has also shed light on its role in other diseases, such as schizophrenia [79], intervertebral disc degeneration [80], gestational diabetes mellitus [81], and rheumatoid arthritis [82], with more exploration required.

## 5. Clinical Applications of miR-574-5p

As noted above, accumulating evidence has revealed the close association between miR-574-5p and various disease developments, especially in multiple cancers, which may support the broad clinical application of miR-574-5p in the future. Some studies have already validated miR-574-5p as a plausible diagnostic or prognostic biomarker, while other reports exploring the underlying molecular mechanisms have suggested its role as a promising therapeutic target, although these studies are in their infancy (Table 3).

### 5.1. Diagnostic Biomarker

Circulating miRNAs have been long recognized as an ideal type of biomarker for their high availability and stability [83]. As they are widely distributed in bodily fluids, such as blood, serum urine, and saliva, and resistant to degradation by endogenous ribonuclease, miRNAs may offer an economical, sensitive, and non-invasive method for human disease diagnosis [84].

Most studies in this field have highlighted the diagnostic value of miR-574-5p in lung cancer, where current tools such as computed tomography (CT) screening still exhibit obvious setbacks [85]. For instance, a report in 2020 found that exosomal miR-574-5p levels varied significantly not only between early-stage lung adenocarcinoma patients and healthy controls, but also between pre- and post-surgery patients. With a combination of miR-342-5p and miR-574-5p, the ROC curve displayed 0.813 (95% CI: 0.7249 to 0.9009) with sensitivity and specificity of 80.0% and 73.2% respectively [86]. Similarly, its role in distinguishing early-stage NSCLC patients has also been revealed, especially with other non-coding RNAs such as hsa-miR-1254 or lncRNA MALAT1 [87,88]. 

In addition to cancer, miR-574-5p also serves as a convenient diagnostic tool in various diseases, such as CAD, thoracic aortic aneurysm (TAA), and even schizophrenia (SCZ) [79,89,90]. Interestingly, as circulating miR-574-5p levels were upregulated in CAD and SCZ patients, they were significantly decreased in TAA tissues but raised in serum samples from another 28 TAA patients compared with controls, a discrepancy perhaps due to the limited statistical power of the study or by the heterogeneity of TAA pathogenesis in patients. This discrepancy also implied that although miR-574-5p has several advantages, its reliability in diagnosis still requires further confirmation.

### 5.2. Prognostic Biomarker

In addition, a growing number of studies have revealed the potential of miR-574-5p in prognosis predictions. For one thing, some reports have demonstrated its association with advanced stages, distant metastasis, or treatment responses in cancers including ESCC, SCLC, and BC [40,91,92]. For another, miR-574-5p also offers the capability of outcome prediction in other diseases. For instance, miR-574-5p had excellent prognostic ability for incident asthma in early childhood with an AUROC of 0.83 under vitamin D level modification [93]. Another study explored its correlation with neurological outcomes after cardiac arrest, suggesting that a higher level of circulating miR-574-5p was associated with poorer neurological outcomes in women (OR [95% CI]: 1.9 [1.09–3.45]), but not in men (OR [95% CI]: 1.0 [0.74–1.28]) [94]. Additionally, two research groups noted its prognostic role in sepsis, despite the seemingly contradictory results. While Liu et al. indicated that down-regulated miR-574-5p was correlated with the onset of acute kidney injury in sepsis patients [95], Wang et al. found that serum miR-574-5p level was associated with death from sepsis, with a significantly higher level in sepsis survivors [96].

Given the inconsistent findings, some other studies noted that its prognostic value was skeptical, at least in certain specific diseases. For example, although the aberrant upregulation of miR-574-5p could serve as a diagnostic biomarker for colorectal cancer, its association with TNM stage or lymph node status was insufficient, mainly because its oncogenic role primarily occurred at the early stages [51]. Similarly, Zhou et al. found that serum miR-574-5p expression was not a predictor for progression of advanced NSCLC, indicating that its role as a prognostic biomarker should be further validated in a larger population [19].

## 6. Conclusions

A large body of literature has accrued on the context-dependent role of miR-574-5p in multiple cancers and other systemic diseases, suggesting its great potential in clinical settings. Functional pathways, networks, and underlying novel mechanisms upregulate or downregulate miR-574-5p, which contribute to tumorigenesis. With the rapid development of nucleic acid therapeutics applied to clinical treatment, miR-574-5p and its related regulatory factors may become potential drug therapeutic targets, providing a valuable breakthrough for cancer treatment and intervention.

In this review, we summarized the intricate functions exerted by miR-574-5p in diverse pathological processes and highlighted the regulatory pathways, networks, and other underlying novel mechanisms. It is noteworthy that although emerging findings have shed light on its therapeutic value as a non-invasive diagnostic and prognostic biomarker, elucidation and further validation are still needed for future clinical translation.

## Figures and Tables

**Figure 1 biomolecules-13-00040-f001:**
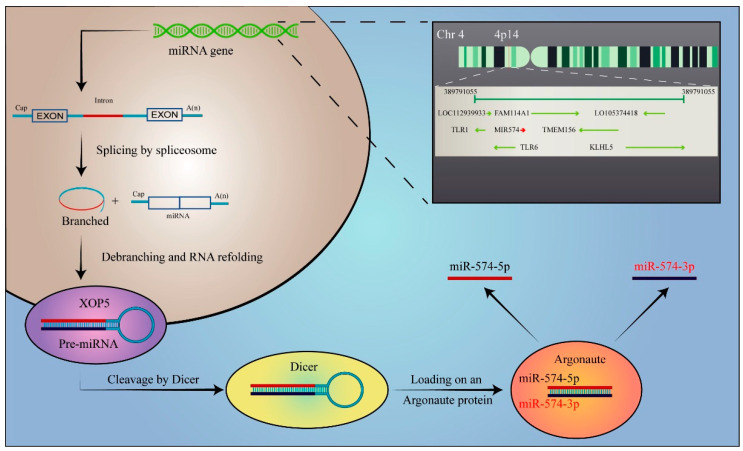
The biogenesis of miR-574-5p. miR-574-5p is first transcribed by RNA polymerase II and mirtrons are obtained. Mirtrons can be spliced and debranched into pre-miRNA hairpin mimics that bypass Drosha cleavage. Debranched mirtrons access the canonical miRNA pathway during nuclear export, and are then cleaved by Dicer and loaded onto argonaute proteins. In the cytoplasm, Dicer endoribonuclease cleaves the ring structure to produce mature miR-574-5p. Human miR-574-5p (has-miR-574-5p) is encoded by mir574 (gene ID: 693159), which is located on the human chromosome 4p14. The sequence and nucleotide sequence of miR-574-5p gene are shown in the figure.

**Figure 2 biomolecules-13-00040-f002:**
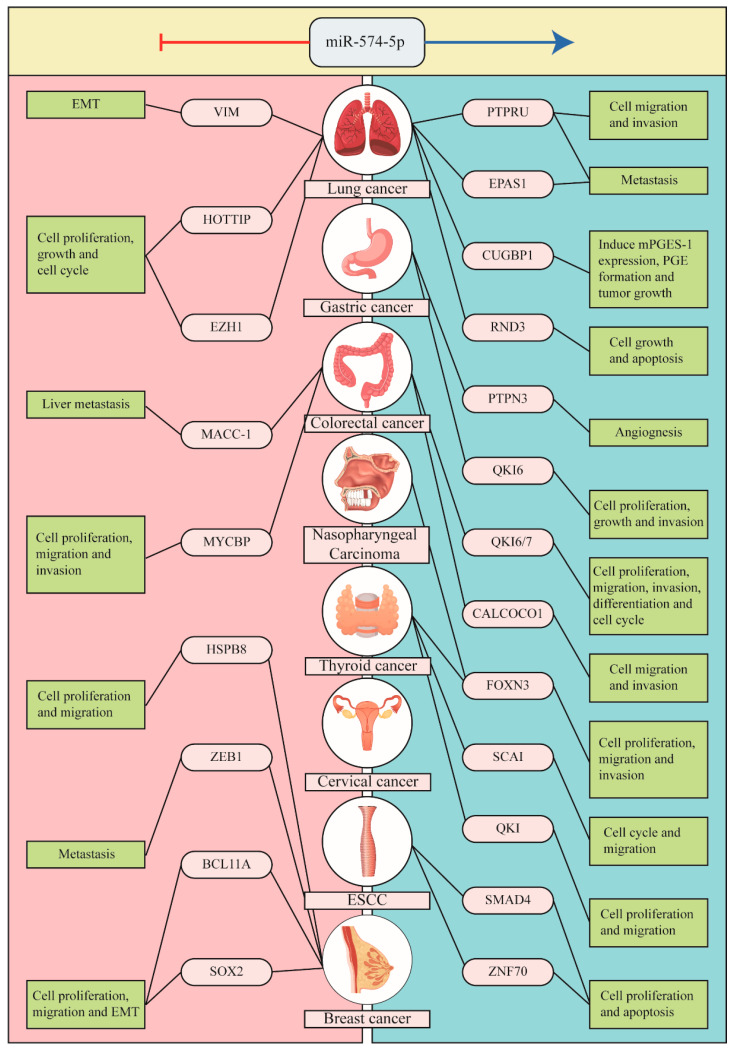
Pathological processes of miR-574-5p in tumorigenesis. Dysregulated miR-574-5p is present in all pathological processes of tumorigenesis, including cancer cell proliferation, migration, invasion, metastasis, apoptosis, epithelial-mesenchymal transition, and angiogenesis. The figure lists some of the roles miR-574-5p plays as a tumor suppressor or as an oncogene in various types of tumors.

**Figure 3 biomolecules-13-00040-f003:**
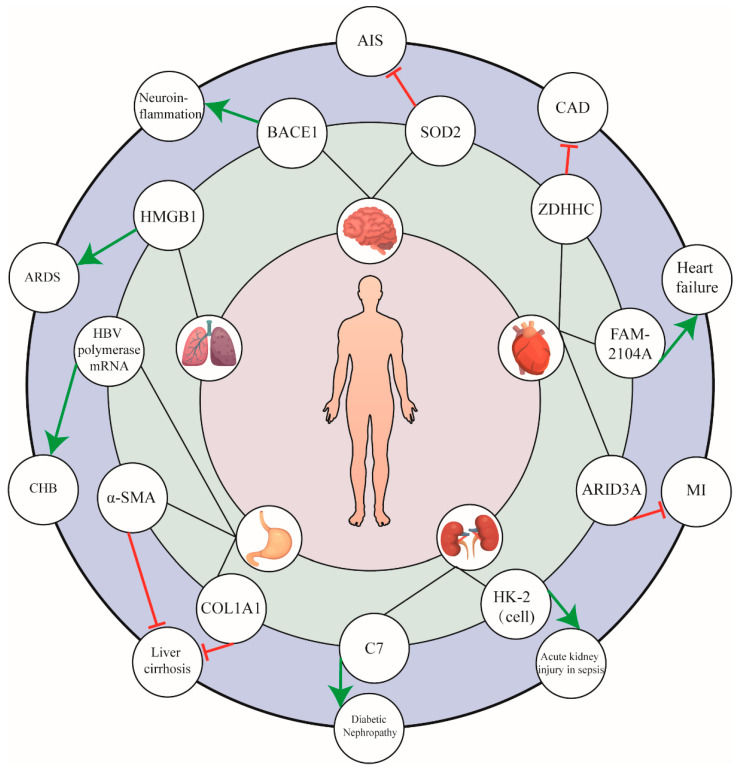
The role of miR-574-5p in non-cancerous diseases. miR-574-5p plays an important role in a wide range of non-cancerous diseases, including cardiovascular, respiratory, acute digestive tract, urinary system, and neurological diseases, among others. The figure illustrates that miR-574-5p directly acts on target cells or indirectly regulates the expression of target genes through signaling pathways, thus promoting or inhibiting the occurrence and development of diseases.

**Table 1 biomolecules-13-00040-t001:** The regulation of miR-574-5p in cancer.

Disease	Pathology	Author	Target	Process	Function	Expression of miR	Impact	Pathways
Lung cancer	small-cell lung cancer (SCLC)	Zhou Rui 2015	PTPRU, EPAS1	metastasis	miR-574-5p significantly enhances the metastasis of SCLC and participates in β-catenin signaling by Suppressing protein tyrosine phosphatase receptor type U (PTPRU) or endothelial PAS domain protein 1 (EPAS1).	↑	↓	Wnt/β-catenin
	non-small cell lung cancer (NSCLC)	Saul et al., 2019	CUGBP1	induce mPGES-1 expression, PGE formation and tumor growth	miR-574-5p induces microsomal prostaglandin E synthase-1 (mPGES-1) expression by preventing CUGBP1 binding to its 3’UTR, leading to enhanced alternative splicing and generation of an mPGES-1 3’UTR isoform, increased mPGES-1 protein expression, and PGE formation	↑	↓	NA
		Emmerich 2020	CUGBP1	induce mPGES-1 expression, PGE formation and tumor growth	miR-574-5p as RNA decoy for CUGBP1 stimulates human lung tumor growth by mPGES-1 induction.	↑	↓	NA
		Zhou Rui 2016	PTPRU	cell migration and invasion	miR-574-5p enhances the tyrosine phosphorylation of β-catenin by repressing PTPRU expression and promotes the migration and invasion of NSCLC cells.	↑	↓	Wnt/β-catenin
		Hua Peng 2016	none	diagnostic biomarker	miR-574-5p can serve as a convenient tool for early NSCLC diagnosis.	↓	NA	NA
		Kristen M. 2011	none	diagnostic biomarker	hsa-miR-574-5p is significantly increased in early-stage NSCLC samples.	↑	NA	NA
	lung adenocarcinoma	Zhijun Han 2020	none	diagnostic biomarker	Circulating exosomal miR-574-5p has potential to serve as novel diagnostic biomarkers for early-stage LA.	↑	NA	NA
Gastric cancer	not-mentioned	Zhengyi Zhang 2019	QKI6	cell proliferation, growth and invasion	The increase in miR-574-5p/-3p ratio, named miR-574 arm-imbalance, is partially due to the dynamic expression of their highly complementary targets in gastric carcinogenesis; the arm-imbalance of miR-574 is in turn involved and further promotes gastric cancer progression.	↑	↓	NA
	not-mentioned	Shu Zhang, 2020	PTPN3	angiogenesis	miR-574-5p in gastric cancer cells promotes angiogenesis via enhancing phosphorylation of p44/42 MAPKs by miR-574-5p inhibition of PTPN3 expression.	↑	↓	MAPKs
	not-mentioned	Jianming Li 2014	none	maintaining the stemness of the gastric CSCs	miR-574-5p shows decreased expression pattern in gastric CSCs	↓	↑	NA
Colorectal cancer	not-mentioned	Shunlong Ji 2012	Qki6/7	cell proliferation, migration, invasion, differentiation and cell cycle	Upregulation of miR-574-5p decreases the expression of Qk6/7 to inhibit b-catenin mRNA and protein expression and inhibit B-catenin /Wnt signal transduction in CRC cells	↑	↓	Wnt/β-catenin
	colorectal cancer liver metastasis	Zhe Cui, 2014	MACC-1	liver metastasis	hsa-miR-574-5p plays a suppressive role in colorectal cancer liver metastasis by negatively directing MACC-1 expression	↓	↑	NA
Esophageal squamous cell carcinoma	not-mentioned	Deqiang Zheng 2019	none	diagnostic and prognostic biomarker	miR574 can distinguish ESCC patients from healthy controls. Moreover, the classifying performance of the miRNA panel can discriminate healthy controls from patients with ESCC stage I-II (AUC > 0.76) and patients with ESCC stage III-IV (AUC > 0.80).	↑	↓	NA
	not-mentioned	MIAO YANG 2013	none	diagnostic biomarker	hsa-miR-574-5p is upregulated in tumor tissues; multiple regression analysis revealed the aberrant expression of hsa-miR-574-5p increases the risk of esophageal cancer.	↑	↓	NA
	not-mentioned	Jie Li 2021	SMAD4	cell proliferation and apoptosis	miR-574-5p represses the abundance of ESCC cells, which leads to booming cell growth of ESCC.	↑	↓	NA
	not-mentioned	Guo Liang Han 2020	ZNF70	cell proliferation and apoptosis	miR-574-5p serve as a tumor promoter regulating cells proliferation and apoptosis in ESCC through and MAPK pathways. Furthermore, ZNF70 has been proven to be a functional target for miR-574-5p to regulate cells proliferation and apoptosis	↑	↓	MAPKs
Breast cancer	triple-negative breast cancer	KE-JING ZHANG 2020	BCL11A, SOX2	cell proliferation, migration and EMT	miR-574-5p attenuates proliferation, migration, and EMT in triple-negative breast cancer cells by targeting BCL11A and SOX2 to inhibit the SKIL/TAZ/CTGF axis.	↓	↑	NA
	not-mentioned	Sheng-kai Huang 2018	none	diagnostic and prognostic biomarker	miR-574-5p might represent a serum biomarker panel with potential in the diagnosis of breast cancer.	↑		NA
	estrogen receptor (ER) positive breast cancer	Margherita Piccolella 2021	HSPB8	cell proliferation and migration	Downregulation of miR-574-5p, which binds to HSPB8 ORF, causes increased expression of HSPB8 protein and the proliferation and migration of ER+BC MCF-7 cells.	↓	↑	NA
Thyroid cancer	not-mentioned	Zhejia Zhang 2018	QKI	cell cycle and migration	miR-574-5p mediates the cell cycle and apoptosis in thyroid cancer cells via Wnt/β-catenin signaling by repressing the expression of Quaking proteins.	↑	↓	Wnt/β-catenin
	not-mentioned	Zhe-JiaZhang 2020	FOXN3	cell migration, proliferation, invasion and apoptosis	miR-574-5p directly targets FOXN3 in thyroid cancer cells, which activates Wnt/β-catenin singling pathway and promotes cell migration, proliferation, invasion and apoptosis	↑	↓(pathway activated)	Wnt/β-catenin
	papillary thyroid carcinoma (PTC)	Xiaoming Wang 2017	SCAI	cell proliferation and migration	PTCSC3 absorbs miR-574-5p, and miR-574-5p targets SCAI; SCAI can regulate the activity of Wnt/β-catenin. PTCSC3/miR-574-5p regulates the activity of Wnt/β-catenin via SCAI and mediates cell proliferation and migration of PTC-1.	↑	↓	Wnt/β-catenin and non-coding RNA
Chordoma	not-mentioned	Emre Can Tuysuza 2019	MYCBP	cell viability, apoptosis, migration and invasion	miR-574-5p targets MYCBP and inhibits the invasive and migratory phenotype; miR-574-5p decreases viability and increases apoptosis in U-CH1 cells while it has no effect on both viability and apoptosis in MUG-Chor1 cells.	↓	↑	NA
Head and neck squamous cell carcinoma	chemotherapy-induced cell death and nodal metastasis	Marisa Meyers-Needham 2011	CerS1-2	cell proliferation and growth	Inhibition of HDAC1 and siRNA-mediated knockdown of miR-574-5p reconstitutes CerS1-2 expression and C18-ceramide generation, which subsequently inhibits cancer cell proliferation.	↑	↓	NA

**Table 2 biomolecules-13-00040-t002:** The regulation of miR-574-5p in other diseases.

Disease Type	Author	Target	Disease	Function	Expression of miR in Disease	Impact	Therapeutic Significance
Cardiovascular diseases	Jianqing Zhou 2016	not mentioned	Coronary artery disease (CAD)	miR-574-5p is significantly upregulated in CAD patients, which indicates that it has great potential to provide sensitive and specific diagnostic value for CAD.	↑	NA	biomarker
	Lai Zhongmeng 2018	ZDHHC	CAD	Upregulation of miR-574-5p enhances cell proliferation and suppresses apoptotic processes in VSMCs through targeting ZDHHC14, suggesting that miR-574-5p is a CAD-related factor that may serve as a potential molecular target for CAD treatment.	↑	↓	potential target
	Wu Jiangbin2021	FAM2104A	Heart failure	miR-574 targets FAM210A and modulates mitochondrial-encoded protein expression, which may contribute to cardiac remodeling in heart failure.	↑	↑	protective role
	CuJun 2020	ARID3A	Cardiac fibrosis after myocardial infarction (MI)	miR-574-5p directly targets ARID3A to promote fibroblast-to-myofibroblast differentiation of TGF-β-induced human cardiac fibroblasts (HCFs)	↑	↓	potential target
	Adeline Boileau 2019	not mentioned	Cardiac arrest	miR-574-5p is associated with neurological outcome after cardiac arrest in women.	↑	NA	biomarker
	Boileau 2019	not mentioned	Thoracic aortic aneurysm (TAA)	miR-574-5p may act as a paracrine mediator in TAA pathogenesis and could be secreted in response to Ang II (further research needed)	↑		biomarker
Respiratory diseases	Binchan He 2020	HMGB1	Acute respiratory distress syndrome (ARDS)	miR-574 targets HMGB1 to inhibit the inflammatory response via suppression of TLR4/NF-Κb signaling pathway and the NLRP3 inflammasome	↑	↑	protective role
	LiJiang 2021	not mentioned	Asthma	hsa-miR-574-5p is a potential mediator and biomarker of asthma and has excellent prognostic power.	↑	NA	biomarker
	LI 2017	not mentioned	Obstructive sleep apnea	miR-574-5p is significantly upregulated in OSA	↑	NA	NA
Digestive system disease	Wu Wenyu 2021	HBV polymerase mRNA	Chronic hepatitis B (CHB)	hsa-miR574-5p downregulates the expression of HBV polymerase mRNA and pgRNA by direct binding to nucleotides 2750-2757	↑	↑	protective role; potential target
	Tan Youwen 2015	not mentioned	Chronic hepatitis B	miR-574-5p could be a surrogate marker for chronic hepatitis B with persistently normal alanine aminotransferase (ALT)	↑	NA	biomarker?
	ZhouXia 2021	α-SMA, COL1A1	Liver cirrhosis	miR-574-5p in serum exosomes transfers to HSC to activate HSC, which is relevant to liver cirrhosis.	↑	↓	potential target
Neurological disorders	Tingting Ku 2017	BACE1	Neuroinflammation	miR-574-5p directly binds to the 3’ UTR of BACE1 to reduce its role of impairing functional synaptic integrity and spatial learning and memory. Overexpression of miR-574-5p in the hippocampal region decreases BACE1 expression, restores synaptic function, and improves spatial memory and learning following PM2.5 exposure.	↓	↑	protective role
	Freischmidt 2013	not mentioned	Amyotrophic lateral sclerosis (ALS)	TDP-43 binding serum miRNA levels are candidates for an easily accessible biological measure of TDP-43 dysfunction in ALS	↓	NA	NA
	Xiaobo Yang 2021	SOD2	Acute ischemic stroke (AIS)	circPHKA2 protects HBMEC from OGD-induced neurovascular injuries by controlling SOD2 via sponging miR-574-5p.	↑	↓	NA
Urinary system diseases	GuoHang 2021	C7	Diabetic Nephropathy	miR-574-5p can negatively regulate expression of C7. Elevated C7 gene expression level in MES is regulated by miR-494-3p and miR-574-5p in early diabetic nephropathy	↓	↑	NA
	Shanshan Liu 2021	HK-2 (cell)	Acute kidney injury in sepsis	Downregulation of miR-574-5p inhibits HK-2 cell viability and predicts the onset of acute kidney injury in sepsis patients.	↓	↑	biomarker
Others	Lip 2020	MKI67	Preeclampsia	miR-574-5p and miR-1972 decrease the proliferation (probably via decreasing MKI67) and/or migration as well as the tube-formation capacity of endothelial cells	↑	↓	potential target
	XuYetao 2020	JDP2	Preeclamptic	miR-574 can directly bind to both AGAP2 AS1 and JDP2, and AGAP2 AS1 regulates JDP2 expression via the AGAP2 AS1/miR 574 axis in trophoblasts	↑	↓	potential target
	Belarbi 2018	EBF1	Obesity/diabetes	miR-574-5p is associated with human adipose morphology and regulates EBF1 expression in white adipose tissue	↑	↓	NA
	WangFuyan 2021	Some predicted targets are associated with the metabolism of glucose and lipids and the insulin signaling pathway.	Gestational diabetes mellitus (GDM)	The expression of miR-574-5p is significantly correlated with levels of blood glucose and LDL-C, which indicates that miR-574-5p may serve as a metabolic regulator of glucose and lipids for GDM.	↓	NA	biomarker
	Davarinejad 2021	not mentioned	Schizophrenia	hsa-miR-574-5P is suggested as a potential biomarker for diagnosis of schizophrenia.	↑	NA	biomarker
	Hegewald 2020	TLR 7/8	Rheumatoid Arthritis	Extracellular miR-574-5p induces osteoclast differentiation via TLR 7/8 in rheumatoid arthritis.	↑	↓	potential target
	Xutao Fan 2021	P3H2	Ontervertebral disc degeneration (IDD)	has-miR-574-5p can suppress the expression on P3H2 in IDD progression	↓	↑	NA
	Wang, Huijuan 2012	not mentioned	Sepsis	serum miR-574-5p is correlated with the death of sepsis patient	↓ in nonsurvivors	NA	biomarker
	Sabetian 2021	ACE2	Male infertility in COVID-19	Downregulation of miR-574-5p may lead to an increase in ACE2 expression related to male infertility in COVID-19	↓	↑	NA

**Table 3 biomolecules-13-00040-t003:** Recent clinical applications of miR-574-5p.

Catagory	Author	Disease	Origin	Sample	Tendency	Diagnosis	Prognosis	Combination
cancer	Zhijun Han 2020	lung adenocarcinoma	serum	7 early-stage lung adenocarcinoma patients including pre-operation and post-operation vs. 7 heathy controls	↑	Yes	Yes	with miR-342-5p
	Zhou Rui 2015	small-cell lung cancer (SCLC)	serum and tissue	a set of 72 SCLC patients (22 limited disease vs. 50 extensive disease)	↑	NA	Yes	none
	Hua Peng 2016	non-small cell lung cancer (NSCLC)	serum	training set: 36 NSCLCs vs. 36 controls validation set: 120 NSCLCs vs. 71 controls	↓	Yes	Yes	a four non-coding RNAs panel (miR-1254, miR-485-5p, miR-574-5p, and MALAT1)
	Kristen M. 2011		serum	training set: 11 early-stage NSCLCs vs. 11 controls validation set: 22 early-stage NSCLCs vs. 31 controls	↑	Yes	NA	with hsa-miR-1254
	Xiao-Rong Yang 2021		plasma exosome	30 EGFR/ALK positive NSCLC patients (16 phase IV patients with bone metastasis and 14 without bone metastasis) vs. 14 healthy donors	↓	NA	Yes	with miR-328-3p and miR-423-3p (all belong to cluster B)
	Sheng-kai Huang 2018	breast cancer	serum	training set: 30 breast cancer patients vs. 30 controls validation set: 128 breast cancer patients vs. 77 controls	↑	Yes	Yes (treatment matters)	A panel of ncRNAs, (let-7a, miR-155, miR-574-5p, and MALAT1)
	Deqiang Zheng 2019	esophageal squamous cell carcinoma (ESCC)	serum	52 ESCC patients vs. 52 age- and sex-matched controls	↑	Yes	Yes	with miR-16-5p, miR-451a
	MIAO YANG 2013		tissue	tumor tissue and non-tumor tissue from 138 ESCC patients	↑	Yes	NA	a set of RNA profiles (hsa-miR-338-3p, hsa-miR-139-5p, hsa-miR-574-5p, and hsa-miR-601)
non-cancer	Jianqing Zhou 2016	coronary artery disease (CAD)	plasma	67 CAD patients vs. 67 healthy controls	↑	Yes	NA	with miR-206
	Adeline Boileau 2019	cardiac arrest	serum	590 cardiac arrest patients	↑	NA	yes (neurological outcome in women)	none
	Boileau 2019	thoracic aortic aneurysm (TAA)	serum and tissue	19 TAA patients and 19 controls; 28 TAA patients compared to 20 controls	↑ in serum and ↓ in tissue	Yes	NA	none
	LiJiang 2021	asthma	plasma	training set: 75 participants with recurrent wheezing at 3 years old from the Vitamin D Antenatal Asthma Reduction Trial validation set: 20 participants in Project Viva with recurrent wheezing at 3 years old	↑	NA	Yes	with vitamin D level; a set of circulating miRNA including hsa-miR-151a-5p
	Shanshan Liu 2021	acute kidney injury in sepsis	serum	136 patients with sepsis (58 developed AKI during hospitalization)	↓	NA	Yes (acute kidney injury)	none
	Wang, Huijuan 2012	sepsis	serum	training set: 12 surviving and 12 nonsurviving sepsis patients validation set: 66 surviving and 52 nonsurviving sepsis patients	↑ in survivors	NA	Yes (death)	with miR-297
	WangFuyan 2021	gestational diabetes mellitus (GDM)	plasma	53 women with GDM and 46 normal pregnant women.	↓	Yes	NA	with miR-3135b
	Davarinejad 2021	Schizophrenia (SCZ)	whole blood	training set: 2 mRNA expression arrays (GSE93987 and GSE38485) and 1 miRNA array (GSE54914) validation set: 40 SCZ patients	↑	Yes	NA	with hsa-miR-1827 and hsa-miR-4429 a

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
