# Peer review of "The Regulatory Mechanism of miR-574-5p Expression in Cancer"

_biomolecules, 2022, doi:10.3390/biom13010040_

Round 1

Reviewer 1 Report

This article by Huang et al. is a comprehensive review of the role of miR-574-5p in cancer. The manuscript is clear and well written and covers the various functions of miRNA in different tumor entities. Overall, this review covers most of the available literature on miR-574-5p, and I appreciate the work done by the authors.

Some minor comments are listed to improve the manuscript:
Figure 1, showing the biosynthesis of miR-574-5p. Please note that this miRNA is processed via a non-canonical miR biogenesis pathway. This miRNA is a so-called mirtron, which is formed from the introns of messenger RNAs (mRNA) during splicing. Please also highlight this in the text.

line 262: “signaling proteins thaare t involved” should be changed to “signaling proteins that are involved”

Line 360: “distributionof” should be changed to “distribution of”

Line 365: 3´UTR isoform not isoforms

Author Response

Response to Editor and Reviewer

Manuscript biomolecules-2075706

Title: " The Regulatory Mechanism of miR-574-5p Expression in Cancer"

Biomolecules

Thank you for handing our manuscript. We greatly appreciate the valuable comments of the reviewers to improve the quality of our manuscript. According to these suggestion and comments, we have made several modifications to our main text and supporting information to ensure readers a better and more accurate reading experience. All the changes has been marked up using the “Track Changes” function in manuscript and a point-by-point response to the editor and reviewers are provided. We hope that our revised manuscript and supporting information will meet the high standard of Biomolecules.

Reviewer1:
1. Figure 1, showing the biosynthesis of miR-574-5p. Please note that this miRNA is processed via a non-canonical miR biogenesis pathway. This miRNA is a so-called mirtron, which is formed from the introns of messenger RNAs (mRNA) during splicing. Please also highlight this in the text.

We appreciate the extraordinary proposal raised by reviewer. We revised the biogenesis pathway of miR-574-5p as “Transcription begins with RNA polymerase II, which produces an initial long transcription chain called mirtrons. Unlike the canonical miR biogenesis pathway, mirtrons can be spliced and debranched into pre-miRNA hairpins that are suitable for Dicer cleavage, thus bypassing the Microprocessor [1-3].” In line 47.

We have also redrawn the relevant steps in the Figure 1, and the figure have been attached separately. Figure legend also accordingly modified as “miR-574-5p is first transcribed by RNA polymerase II and mirtrons is obtained. Mirtrons can be spliced and debranched into pre-miRNA hairpin mimics that bypass Drosha cleavage. Debranched mirtrons access the canonical miRNA pathway during nuclear export, and are then cleaved by Dicer and loaded onto Argonaute proteins.” in line 73.

  1. line 262: “signaling proteins thaare t involved” should be changed to “signaling proteins that are involved”

We are grateful for the professional view of the reviewer and sorry for the mistake. We revised the wrong spellings as “signaling proteins that are involved in many basic physiological processes”.

  1. Line 360: “distributionof” should be changed to “distribution of”

We are sorry for the mistake, and we have corrected the wrong spellings in line 369.

  1. Line 365: 3´UTR isoform not isoforms

    We are thankful for the valuable point of the reviewer and have revised the mistake in line 374.

Reference:

  1. Ha M, Kim V. Regulation of microRNA biogenesis. Nature reviews Molecular cell biology. 2014;15(8):509-24.
  2. Han J, Lee Y, Yeom K, Kim Y, Jin H, Kim V. The Drosha-DGCR8 complex in primary microRNA processing. Genes & development. 2004;18(24):3016-27.
  3. Havens MA, Reich AA, Duelli DM, Hastings ML. Biogenesis of mammalian microRNAs by a non-canonical processing pathway. Nucleic Acids Res. 2012;40(10):4626-40.

Reviewer 2 Report

The review is updated and comprehensive

I would suggest to eliminate the sections 3.2.2 circ RNAs since the whole field is pretty unclear and it does not add much to the review itself.
I would de emphasized the ceRNA focus since the role of lncRNAS and other non-coding species in sponging miRNAs is not completely accepted in the field if not for limited RNAs that have over 70 MREs for the same miRNA. Ref 64 and 65 are controversial, based on bioinformatic analysis and very poor experimental data, never confirmed, lack of follow up studies from the original lab, authors should consider to remove them
The last section 5.3 about therapeutic application can be eliminated since there are no preclinical nor clinical evidence that miR-574 can be used as therapeutic target and there are no programs trying to develop any anti-miR nor mimic based therapy for this miRNA.

Table can be formatted in a pdf like format and put all together in supplementary to make the whole manuscript easy to read

Overall interesting, need minor editing and can be published

Author Response

Reviewer2:

  1. I would suggest to eliminate the sections 3.2.2 circ RNAs since the whole field is pretty unclear and it does not add much to the review itself.

We appreciate the great proposal of the reviewer. We realized that the current research in this field is still controversial and does not provide a substantial supplement to the overall framework of the review. We have removed the relevant content from the manuscript.

  1. I would emphasized the ceRNA focus since the role of lncRNAS and other non-coding species in sponging miRNAs is not completely accepted in the field if not for limited RNAs that have over 70 MREs for the same miRNA. Ref 64 and 65 are controversial, based on bioinformatic analysis and very poor experimental data, never confirmed, lack of follow up studies from the original lab, authors should consider to remove them.

We recognized that the prerequisite for lncRNAs and other non-coding species to play a role in sponging miRNAs is very strict, and some experimental data cannot strongly and directly prove our conjectures. In order to further strengthen the rigor of the review and consider the logical integrity of the article, section 3.2 was deleted and the text related to the role of IncRNA and other non-coding species mentioned in the previous text was corrected.

  1. The last section 5.3 about therapeutic application can be eliminated since there are no preclinical nor clinical evidence that miR-574 can be used as therapeutic target and there are no programs trying to develop any anti-miR nor mimic based therapy for this miRNA.

We appreciate the professional view of the reviewer. We further searched the clinical research reports in recent years, and found that the field is still in a large blank, and there is no strong evidence to confirm our expectation at present. We agreed with your suggestion and have deleted this section from the manuscript.

  1. Table can be formatted in a pdf like format and put all together in supplementary to make the whole manuscript easy to read

Thanks for the reviewer's professional advice, and sorry for bringing a bad reading experience. We converted the form into PDF format and uploaded it separately, hoping to facilitate the reviewer's reading.
